# Severe Quadriceps Heterotopic Ossification after Knee Revision Arthroplasty in a 42-Year-Old Suffering from Rheumatoid Arthritis: A Case Report

**Michele Massaro [1], Federico Mela [1], Roberto Esposito [1], Emanuele Maiorano [1] and Guy Laskow [1,2,*]**

1   Minimally Invasive Advanced Robotic Prosthetic Orthopedics Unit, OPARM—Gruppo Ospedaliero San Donato, via Forlanini n. 15, Ponte S. Pietro, 24036 Bergamo, Italy
2   Orthopedic Residency Program, Vita-Salute San Raffaele University, Via Olgettina n. 60, 20132 Milan, Italy
*   Correspondence: dr.laskow@gmail.com; Tel.: +39-345-3097399

**Abstract:** Background: Heterotopic Ossification (HO) of the knee is most commonly formed anteriorly to the distal femoral shaft in the quadriceps expansion. Although the incidence of severe HO with large dimensions affecting the knee and resulting in severe consequences is extremely rare, these cases are extremely difficult to prevent and have severe clinical limitations for the patient. Aim: The purpose of this study was to present and explore HO formation after Total Knee Arthroplasty (TKA). Conclusions: It is crucial to perform a stratification of patients for the risk of HO formation after TKA and to gain a better understanding of the fundamental role of post-operative treatments. In severe HO, surgery should be considered following appropriate investigations and should only be considered when the HO has fully matured. In comparison to Total Hip Arthroplasty (THA), HO formation after TKA is less frequent and underexplored. Therefore, further studies are required. This case report can represent a protocol for the treatment of clinically relevant HO in the knee after TKA.

**Keywords:** quadriceps heterotopic ossification; unicompartmental knee arthroplasty; revision





## 1. Introduction

We present a case of severe quadriceps Heterotopic Ossification (HO) following a revision of a Unicompartmental Knee Arthroplasty (UKA) with a Total Knee Arthroplasty (TKA) in a patient with Rheumatoid Arthritis, and the subsequent treatment and efficacy of Prophylactic Radiotherapy (PRT) as a secondary prevention following surgical resection of clinically apparent HO.

The overall incidence of HO after TKA is about 15% [1]. Although this incidence is lower than the most common HO after Total Hip Arthroplasty (THA), which varies between 4% and 42% reported [2], HO after TKA is thought to be underreported. The most common complaint related to the development of HO is stiffness; in cases of large HO, the stiffness could progressively lead to an increasing limitation of range of motion (ROM) or even ankylosis in the affected joint.

HO of the knee is most commonly formed anterior to the distal femoral shaft in the quadriceps expansion [1]. Although the incidence of severe HO with large dimensions affecting the knee and resulting in these severe consequences is extremely rare and confined to a few case reports, these cases are extremely difficult to prevent and have severe clinical limitations for the patient.

Cases of clinically relevant HO can be treated with radiation therapy, surgical excision of the HO, manipulation of the affected knee joint under anaesthesia, and possibly revision of the arthroplasty components [1]. Patients with Rheumatoid Arthritis showed less HO, which is not a clear risk factor for the development of clinically relevant HO formation [3].

## 2. Case Presentation

A 42-year-old woman with a Body Mass Index (BMI) of 27.2 and a background of Rheumatoid Arthritis, who had undergone THA 3 years previously (with previous incidences of Heterotopic Ossification treated successfully with radiotherapy) and UKA one year previously, came to our clinic because of a limitation of knee ROM and disabling pain. After an initial visit that included an X-ray, as shown in Figure 1A, a clinical evaluation and clinical tests, including a blood test, were performed to rule out underlying infections. We then decided to revise the UKA with a Cruciate-Retaining Total Knee Arthroplasty (CR-TKA), as shown in Figure 1B.

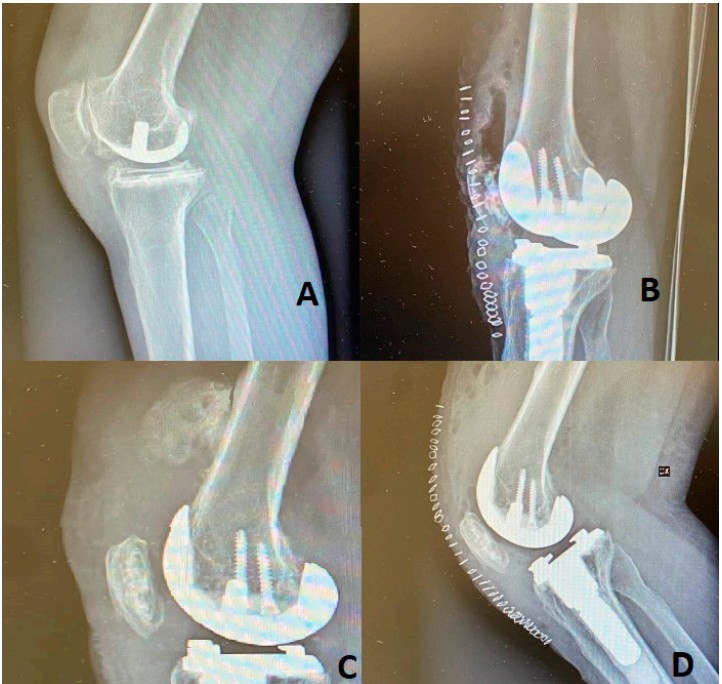

**Figure 1.** (**A**): UKA radiography of the patient's knee. (**B**): X-ray of the Revision of UKA with CR-TKA. (**C**): Radiography demonstrating the HO. (**D**): Post operative X-ray after HO removal.

The standard medial para-patellar incision with a mini mid-vastus approach was performed. The surgery was performed by the head of our unit, who is an arthroplasty surgeon with more than fifteen years of experience. The preoperative range of motion in the knee was 10-90 degrees, which was increased to 0-110 degrees postoperatively. The patient underwent postoperative X-ray, as shown in Figure 1B, and rehabilitation consisting of proprioceptive training, neuromuscular reeducation, reinforcement of the thigh muscles, and Continuous Passive Motion (CPM) with a Kinetec device in our clinic. The patient was discharged five days postoperatively. It was not possible to continue the physiotherapy protocol in the outpatient clinic due to the Sars-Cov-2 pandemic, resulting in a failure to regain full extension or flexion of the knee. At the two-month follow-up, the patient came to our clinic with radiography, which demonstrated a large calcification in the distal femoral shaft in the quadriceps expansion as shown in Figure 1C. At that time, ROM was 5–60 degrees. Therefore, we decided to restart physiotherapy. At the six-month follow-up, after three months of physiotherapy, knee ROM was not improving, so it was decided to surgically treat the patient to remove the HO that measured 44 mm, as shown in Figure 2. After the surgical excision, knee ROM was 0–100 degrees. The patient maintained a 0-90-degree ROM and radiography equivalent to the post-surgical that showed no calcifications at the nine-month post-surgery follow-up, as shown in Figure 1D.

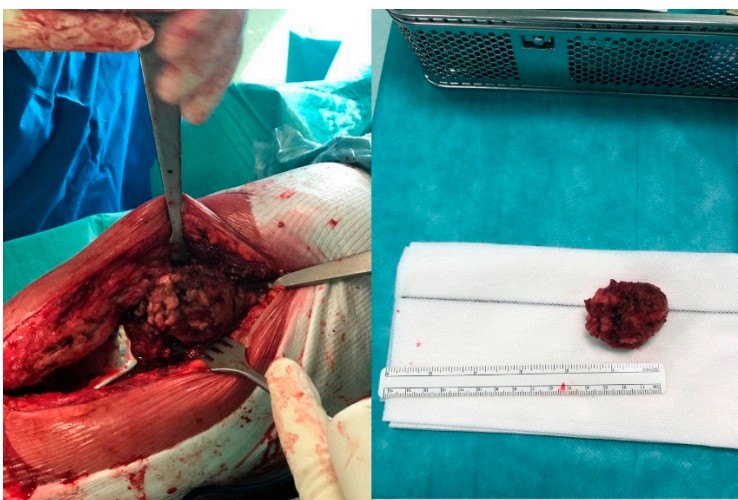

**Figure 2.** Surgical removal of the Heterotopic Ossification.

Thanks to the experience gained with hip HO, which had been successfully treated by postoperative radiotherapy [4], it was decided that the patient would undergo three fractions of seven Gray units (Gy) per fraction of PRT sessions to the knee. In addition, we prescribed intense physiotherapy immediately post-operatively, which the patient conducted for a further two months.

## 3. Discussion

HO of the knee consists of idiopathic bone formation in soft-tissue structures, which can cause nerve entrapment, joint dysfunction, or ankylosis. It is mostly observed following trauma or surgery, but also has other, less common causes such as neurological injury or burns. There is some disagreement among studies regarding gender as a risk factor; some point to male gender as a risk group, while others point to female [2]. Individual risk factors include genetic predisposition and obesity. The established facts are that high-risk factors include pre-existing or contralateral HO, hypertrophic osteoarthritis, ankylosing spondylitis, diffuse idiopathic skeletal hyperostosis, infection, previous injuries to the knee, rheumatoid arthritis, early septic arthritis, preoperative knee deformity, and increased lumbar bone marrow density [5], excessive manipulation and extensive soft tissue dissection during surgery, such as quadriceps tendon splitting, stripping of the anterior femur for measuring purposes, chronic knee effusions, notching of the femur, vigorous soft tissue retraction, hematoma formation, retained bone particles from bone resections, and "press-fit" fixation surgical technique of the tibia. The HO formation mechanism is still poorly known, yet studies show that the main driver is the mesenchymal stem, which causes the inflammatory state that is correlated with HO formation. Other inflammatory mediators also help with HO formation, such as prostaglandins and fibroblast growth factors [6]. The main symptoms for clinically significant HO are the following: severe pain and discomfort that may lead to loss of function, limited ambulation, and decreased range of motion that could cause stiffness at the knee joint [1]. Other less common symptoms are swelling or warmth in the joint area, fever, and increased spasticity [7]. Most of the HO formation is clinically irrelevant; clinically significant HO of the knee was reported in roughly 20% of cases. According to studies, postoperative ROM improved by 82% after surgical debulking of the knee. Patients also had improved ambulation in 57% of cases and improved sitting ability in 93% of cases [8]. HO formation is typically first seen at 4 weeks postoperatively in plain radiographs either for asymptomatic or symptomatic patients. Studies show that examination of patients at 1 year demonstrated no effect of HO on the range of motion. The extent of HO is noted to stabilise at 1 year, and HO even resolved spontaneously in several patients. A novel insight into HO formation in the knee joint is that it follows a distinctly different process from that observed in the hip joint. HO of

the knee is most often observed initially either in the periarticular soft tissues or along the anterior edge of the distal femur. Deposits of HO are frequently observed in the medial aspect of the knee joint in the area known as the quadriceps expansion (6). There are five different classification systems for HO in the knee joint (by Harwin, Dalury, Furia, Rader, and Toyoda); these classifications are not uniform, as the clinical aspect, location and size are not conclusive with each other. The assessment of TKA severity is unreliable in the absence of a single, comprehensive, standardized classification system [9]. Prevention of HO in the knee after TKA is still inconclusive, and studies based on HO formatting in the hip after THA showed that selective COX-2 Inhibitors and non-selective NSAIDs such as Diclofenac and Indomethacin are equally effective in the prevention of HO, but have inferior results to Etoricoxib if prescribed for 10 days after surgery [10]. Perioperative radiation is also a prophylactic measure, but it is only effective when performed in a time window of 20 h before and 96 h after surgery (optimal effect 8 h before and 72 h after surgery) [11].

## 4. Conclusions

It is crucial to perform a stratification of patients for risk of HO formation after TKA and to have a better understanding of the fundamental rule of physiotherapy and of the risks involved in HO formation. In severe HO, surgery should be performed following appropriate investigations and should only be considered when the HO has fully matured. HO formation post TKA is less frequent than HO formation post-THA, but that could be because it remains an underexplored argument compared to HO post-THA, and further studies are required. This case report can represent a protocol for the treatment of clinically relevant HO in the knee after TKA, but further research is needed.

**Author Contributions:** Conceptualization, M.M.; methodology, M.M.; software, M.M.; validation, M.M., R.E.; formal analysis, G.L.; investigation, G.L.; resources, E.M.; data curation, F.M.; writing—original draft preparation, F.M.; writing—review and editing, G.L.; visualization, G.L.; supervision, M.M.; project administration, G.L.; funding acquisition, M.M All authors have read and agreed to the published version of the manuscript.

**Funding:** This research received no external funding.

**Institutional Review Board Statement:** The study was conducted according to the Declaration of Helsinki concerning medical research, and written informed consent was obtained from the patient.

**Informed Consent Statement:** Informed consent was obtained from the subject involved in the study. Written informed consent has been obtained from the patient to publish this paper.

**Data Availability Statement:** All data generated or analyzed during this study are included in this published article. We do not wish to share our patients' data because it involves the patient's privacy.

**Acknowledgments:** The authors thank Jessica Ben Haim and Katia Pasta for their valuable assistance and support.

**Conflicts of Interest:** The authors declare no conflict of interest.

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
