# Peer review of "Severe Quadriceps Heterotopic Ossification after Knee Revision Arthroplasty in a 42-Year-Old Suffering from Rheumatoid Arthritis: A Case Report"

_2673-4036, doi:10.3390/osteology2040019_

Round 1

Reviewer 1 Report

Comments and Suggestions for Authors

This is an interesting case report that highlights the gaps in the protocols enforced, which ultimately shows that the burden of efficient treatment falls on timely decisions and the experience of the attending.

Kindly consider a better presentation of the case in terms of the timetable. It is essential to have a clear view of the patient's progress in relation to the treatments.  It might be helpful for the authors to split the case presentation into the relevant time points and provide a detailed description of their observations, examination findings, and treatment decisions at each time.

Also please be sure to revisit the case presentation text for minor errors ( lines 64-66).

The results section still has the template's text, so please remove that and replace it appropriately (it might be useful to provide the examination/treatment outcomes for the patient here). Also if there is any other baseline information about the patient (other than gender and age) that might be useful for the reader the authors may consider including it as well.

Author Response

Dear Reviewer,

Thank you so much for the help and constructive feedback, I’ve made the required changes and added more information about the timetable, fixing minor errors, removing the result section, adding the BMI of the patient, and adding X-rays of the patient.

Good day

Dr. Guy Laskow

Reviewer 2 Report

Comments and Suggestions for Authors

1. Please specify any acronyms when they are mentioned for the first time. For example, HO in the abstract.

2. Please carefully edit the language and correct typographical errors. For eg. in the abstract, o fthis, etc.

3. Please remove the results section as it is not applicable.

4. Was the calcification not observed during the imaging investigations? It would be better to provide images as Figures.

5. Similarly, provide the histopathological diagnosis and image.

Author Response

Dear Reviewer,

Thank you so much for the help and constructive feedback, I’ve made the required changes and added more information about the timetable, fixing minor errors, removing the result section, adding the BMI of the patient, and adding X-rays of the patient.

Unfortunately histopathological diagnosis was not preformed 

Good day

Dr. Guy Laskow

Round 2

Reviewer 1 Report

Comments and Suggestions for Authors

The authors responded to all comments 

Reviewer 2 Report

Comments and Suggestions for Authors

None.